# The Extraction Solvent Influences the Anti-Inflammatory Effects of Jakyakgamcho-Tang in Lipopolysaccharide-Stimulated Macrophages and Mice with Gouty Arthritis

**DOI:** 10.3390/ijms21249748

**Published:** 2020-12-21

**Authors:** Yun Mi Lee, Dong-Seon Kim

**Affiliations:** Herbal Medicine Research Division, Korea Institute of Oriental Medicine, 1672 Yuseongdae-ro, Yuseong-gu, Daejeon 34054, Korea; candykong@kiom.re.kr

**Keywords:** Jakyakgamcho-Tang, traditional medicine, inflammation, macrophages, gouty arthritis

## Abstract

Jakyakgamcho-Tang (JGT) is a traditional medicine used to treat muscular tension, spasm, and pain. Several studies have reported its clinical use as an anti-inflammatory and in gynaecological treatment. This study aimed to compare the anti-inflammatory effects of JGT according to extraction solvent, water (JGTW) and 30% EtOH (JGTE) on lipopolysaccharide (LPS)—stimulated macrophages and in mice with monosodium urate (MSU)—induced gouty arthritis. We evaluated the production of inflammatory mediators and cytokines and the expression of inducible nitric oxide (iNOS) and cyclooxygenase-2 (COX-2) in RAW 264.7 cells. We also examined oedema, pain, and inflammation in MSU-induced mice by measuring affected hind paw swelling, weight-bearing, pro-inflammatory cytokines levels, and myeloperoxidase (MPO) activity. In LPS-stimulated RAW264.7 cells, JGTW and JGTE significantly decreased prostaglandin (PG) E2(PGE2) production via suppressing COX-2 expression and cytokines interleukin-1β and interleukin-6. Only JGTE reduced the production of NO and cytokines and the mRNA levels of iNOS and cytokines. In MSU-induced mice, JGTE and JGTW efficiently decreased paw swelling and attenuated joint pain. JGTE (200 and 300 mg/kg) effectively suppressed inflammation by downregulating pro-inflammatory cytokines (tumour necrosis factor (TNF)-α, interleukin (IL)-1β, and IL-6) and MPO activity, which were only slightly reduced by JGTW. Our data demonstrate the anti-inflammatory activity of JGT in macrophage and gouty arthritis animal models and show that JGTE is more effective than JGTW at lower concentrations.

## 1. Introduction

Jakyakgamcho-Tang (JGT; Shakuyaku-kanzo-to in Japanese; Shaoyao-gancao-tang in Chinese) is a traditional herbal medicine composed of *Paeonia lactiflora* Pall. (Paeoniae Radix) and *Glycyrrhiza uralensis* Fisch (Glycyrrhizae Radix et Rhizoma). JGT has been clinically and pharmacologically used in Korea and Japan for the treatment of acute abdominal pain, muscle pain, backache, neuralgia, painful peripheral neuropathy, and bronchial asthma [1,2,3,4,5]. Other applications of JGT, such as gynaecological therapeutics and anti-inflammatory treatment, are now emerging as new areas in its clinical use [6]. We have previously reported that the major constituents in JGT are oxypaeoniflorin, paeoniflorin, pentagalloylglucose, benzoyl paeoniflorin, liquiritin apioside, liquiritin, isoliquiritin, apioside, liquiritigenin, and glycyrrhizin, and that the relative amounts of JGT components are approximately 19–53% higher in the 30% ethanol extract than in the water extract. The 30% ethanol extract of JGT improves the exhaustion swimming time (54%) and levels of serum lactate dehydrogenase (48%) and lactic acid (60%), confirming an anti-fatigue effect in forced swimming ICR mice [7]. Although the medicinal properties of JGT have been reported in several studies, the impact of the extraction solvent on the anti-inflammatory action of JGT in lipopolysaccharide (LPS)-stimulated macrophages and monosodium urate (MSU)-induced gouty arthritis models has not been investigated yet. LPS-induced RAW 264.7 macrophages are a widely used model to study inflammatory responses in vitro [8]. Macrophages activated by a variety of inflammatory mediators, such as nitric oxide (NO), prostaglandin (PG) E2, cyclooxygenase-2 (COX-2), inducible expression of nitric oxide synthase (iNOS), 5-lipoxygenase (5-LOX), tumour necrosis factor (TNF)-α, interleukin (IL)-1β, and IL-6, contribute to the pathogenesis of inflammatory diseases. Therefore, inhibiting the production of these pro-inflammatory mediators could be a highly efficient tool for blocking the development and progression of inflammatory diseases [9].

Gouty arthritis, which affects a growing number of people, is an inflammatory joint disorder caused by the deposition of MSU in the joints, leading to an intense inflammatory response and pain [10]. The patients present with redness, swelling, and severe pain in the affected joints and surrounding tissues [11]. Non-steroidal anti-inflammatory drugs (NSAIDs) and colchicine are recommended first-line treatment options for acute attacks of gouty arthritis [12]. Although these agents are generally effective, they also present adverse effects in patients with pre-existing gastrointestinal, renal, and cardiovascular diseases [13]. In addition, colchicine has a narrow therapeutic index and is linked to a relatively uncommon but fatal toxic reaction [14]. Therefore, the current research on the development of medicines for gouty arthritis has focused on natural products with good efficacy and low adverse effects [15,16]. Recently, ethanol or ethanol/water mixtures have been used as extraction solvents for pharmaceuticals and dietary supplements. The Korea Food and Drug Administration exempts or requires minimum toxicity test data for drug approval of oriental herbal medicine when using ethanol content up to 30% in mixture with water as an extraction solvent. Thus, this study investigated the efficacy of JGT according to two extraction solvents, water (JGTW) and 30% EtOH (JGTE). The anti-inflammatory activities of the two types of JGT extract were compared in LPS-treated macrophages and an MSU-induced gouty arthritis mouse model.

## 2. Results

### 2.1. JGT Inhibits Activity of 5-LOX and COX Enzymes

First, we tested the inhibitory effects of JGTW and JGTE on 5-LOX and COX activity. As shown in Table 1, JGTE inhibited 5-LOX enzyme activity with an IC_50_ of 74.63 μg/mL, while JGTW was inactive (IC_50_ > 500 μg/mL) towards this enzyme. In addition, JGTW suppressed COX-1 and COX-2 enzyme activity in a dose-dependent manner with IC_50_ of 219.47 μg/mL and 348.47 μg/mL, respectively. JGTE also dose-dependently reduced COX-1 and COX-2 enzyme activity with IC_50_ of 97.77 μg/mL and 114.16 μg/mL, respectively.

### 2.2. JGTE Reduces NO/PGE2 Production and iNOS/COX-2 Expression More Effectively Than JGTW in LPS-Induced RAW 264.7 Cells

Next, we examined the toxicity of several concentrations of JGTW and JGTE in RAW 264.7 cells. After 24 h of treatment, the cell viability was not affected up to 200 μg/mL of both JGTW and JGTE (Figure 1A). As shown in Figure 1B, NO production was significantly higher in LPS-treated cells compared to untreated control (Con) cells; however, pre-treatment with JGTE significantly reduced NO production by 19.96% and 75.22% at doses of 100 and 200 µg/mL, respectively. Instead, JGTW did not significantly affect NO production in LPS-induced RAW264.7 cells. Additionally, both JGTW and JGTE significantly reduced PGE2 production in a dose-dependent manner, but the inhibitory effect of 50 μg/mL JGTE was more effective than that of 100 μg/mL JGTW (Figure 1C). Next, we measured the mRNA levels of iNOS and COX-2 using real-time PCR. The expression of iNOS and COX-2 was markedly higher in LPS-stimulated cells than in Con cells (Figure 1D,E). JGTW slightly decreased, although not significantly, the expression of iNOS and significantly reduced that of COX-2 in LPS-treated RAW264.7 cells. Compared with JGTW, JGTE reduced the expression of iNOS and COX-2 in LPS-stimulated cells more effectively, in a dose-dependent manner (Figure 1D,E).

### 2.3. JGTE Has a Stronger Inhibitory Effect Than JGTW on Inflammatory Cytokine Production in LPS-Induced RAW 264.7 Cells

To investigate the inhibitory effects of JGTW and JGTE on pro-inflammatory cytokine production and gene expression, we measured the production and mRNA levels of TNF-α, IL-1β, and IL-6 in LPS-stimulated RAW 264.7 cells. As shown in Figure 2A,B, LPS treatment significantly increased TNF-α production and gene expression, whereas pre-treatment with 100 and 200 µg/mL JGTE significantly downregulated TNF-α production by 22.5% and 27.1%, and TNF-α mRNA expression by 43.6% and 58.9%, respectively. However, JGTW did not show any significant effects on TNF-α production and gene expression in LPS-induced RAW264.7 cells. Regarding IL-6 production, the inhibitory effect of JGTE was 7.6- and 5.4-fold stronger than that of JGTW at the same concentrations of 100 μg/mL and 200 μg/mL, respectively (Figure 2C,D). In addition, both JGTW and JGTE significantly reduced IL-1β production in a concentration-dependent manner; however, the inhibitory effect of JGTW at doses of 100 μg/mL and 200 μg/mL was comparable to that of JGTE at doses of 50 μg/mL and 100 μg/mL (Figure 2E,F).

### 2.4. JGTE Shows a Stronger Anti-Inflammatory Effect Than JGTW on MSU-Induced Paw Oedema

MSU crystals led to a significant increase in the paw thickness of injected mice (Figure 3A,B). However, JGTW (200 and 300 mg/kg) and JGTE (100, 200, and 300 mg/kg) markedly decreased the MSU-induced paw oedema by 8.1%, 20.2%, 21.4%, 26.0%, and 26.1%, respectively.

At the same concentrations, the JGTE group showed a greater change in paw thickness than the JGTW group. Moreover, JGTE at concentrations of 100 mg/kg and 200 mg/kg was 2.6- and 3.2-fold more effective, respectively, than JGTW at 200 mg/kg in contrasting paw oedema.

### 2.5. JGT Restores Hind Paw Weight-Bearing Distribution in MSU-Injected Mice

To assess the progressive pain of gouty arthritis, we measured the ratio of hind paw weight distribution between the right and left paw. The weight-bearing distribution was remarkably lower in the MSU crystal group than in the control group mice. By contrast, DWB measurements showed that JGTW, JGTE, and Col-treated groups displayed a significant reversal of this MSU crystal-induced inflammatory pain (Figure 4). Hind paw weight distribution was significantly elevated in JGTW (300 mg/kg), JGTE (200 mg/kg), and Col treatment groups to similar levels. In particular, JGTE at the dose of 300 mg/kg was the most effective in restoring weight distribution in MSU crystal-induced mice.

### 2.6. JGT Significantly Reduces Proinflammatory Cytokine Levels in MSU-Injected Mice

The levels of IL-1β, IL-6, and TNF-α were examined in JGTW- and JGTE-treated mice after the injection of MSU crystals. As shown in Figure 5, MSU-injected mice had significantly increased IL-1β, IL-6, and TNF-α levels, but treatment with JGTE significantly downregulated the production of these pro-inflammatory cytokines. Instead, in JGTW-treated mice, the levels of IL-1β, IL-6, and TNF-α were slightly reduced, although not significantly.

### 2.7. JGT Restrains MPO Activity and Thereby Inhibits Inflammation

To evaluate the possible cellular infiltration induced by MSU, we examined myeloperoxidase (MPO) activity as an index of neutrophil accumulation. As shown in Figure 6, the injection of MSU drastically increased (*p* < 0.0001) MPO activity in the paw tissue. JGTW and JGTE (300 mg/kg) reduced the activity of MPO, with the highest effect observed with JGTE at a dose of 300 mg/kg (*p* < 0.01). The positive control Col (1 mg/kg), which inhibits neutrophil recruitment and activation, also significantly reduced MPO activity (*p* < 0.05). Therefore, JGTE could restrain MPO activity and effectively inhibit inflammation.

## 3. Discussion

Macrophages play an important anti-inflammatory role and can decrease immune reactions through the release of cytokines [17]. Activation of macrophages by LPS causes the production of inflammatory mediators (NO, iNOS, PGE2, and COX-2) and pro-inflammatory cytokines (TNF-α, IL-1β, and IL-6), which contribute to the progression of several inflammatory diseases [18,19]. NO and PGE2, which are produced by iNOS and COX-2, respectively, also modulate inflammation and pain. In the present study, to investigate the effects of the extraction solvent on JGT during the inflammatory process, RAW264.7 macrophage cells were stimulated with LPS. Both JGTW and JGTE efficiently reduced the production of PGE2 via suppression of COX-2 expression, but the inhibitory effect on NO production was confirmed only for JGTE. In LPS-stimulated RAW264.7 cells, JGTE reduced NO production in a concentration-dependent manner, which correlated well with the dose-dependent decrease in iNOS mRNA levels (Figure 1).

COX-1, COX-2, and 5-LOX, key pro-inflammatory enzymes, as well as several cytokines are associated with promoting inflammation. [20]. Here, JGTE showed an IC_50_ value of 74.63 μg/mL for 5-LOX, while JGTW showed < 50% inhibition of the above enzymes at the highest concentration tested (500 μg/mL). In addition, the IC_50_ of JGTE for COX-1 and COX-2 (97.77 μg/mL and 114.16 μg/mL) was 2.24-fold and 3.05-fold lower than that of JGTW (219.47 μg/mL and 348.47 μg/mL), respectively, suggesting JGTE as a more effective inhibitor. Moreover, JGTE significantly reduced the secretion and mRNA expression of IL-1β, IL-6, and TNF-α in a concentration-dependent manner. Instead, JGTW produced a slight, dose-dependent decrease in IL-1β, IL-6, and TNF-α production and mRNA gene expression, but these results were only statistically significant for the production of IL-1β and IL-6.

Gouty arthritis is characterised by an intense inflammatory process and pain caused by the deposition of MSU crystals within the joints and connective tissues [21]. MSU crystals activate the secretion of inflammatory cytokines, including IL-1β, TNF-α, and IL-6 by macrophages, neutrophils, and monocytes-macrophages [22]. Therefore, suppressing MSU-induced neutrophil recruitment and blocking inflammation-mediated cytokine secretion to reduce swelling, pain, and inflammation might be beneficial for the control and management of acute gouty arthritis [23,24]. In the present study, MSU-induced mice showed a significant increase in swelling and markedly reduced weight-bearing on the affected hind paw, indicating pain. The 200 mg/kg JGTE dose was more effective than the 300 mg/kg JGTW dose and had a similar or slightly better inhibitory effect than the Col positive control. In addition, the levels of IL-1β, IL-6, and TNF-α in the paw tissue, which were significantly increased in response to MSU, were dose-dependently downregulated by JGTE treatment; the effect of JGTW was not statistically significant. Furthermore, MPO activity was significantly higher in mice with gouty arthritis than in the control group (indicating an influx of neutrophils and acute inflammation), and both JGTW and JGTE induced a decrease in the activity of this enzyme. Again, treatment with the 300 mg/kg JGTW dose showed similar efficacy to that of 200 mg/kg JGTE and colchicine, which is a known regulator of neutrophil activity [25]; at the same concentration, JGTE treatment efficacy was superior compared to JGTW treatment. These results suggest that JGTE relieves pain and swelling, acute gout symptoms caused by MSU crystals by inhibiting the key inflammatory cytokines and MPO activity, which is a key feature in the initiation and progression of gouty arthritis. Furthermore, our data indicate that JGTE treatment at the same concentration was more effective than JGTW.

Extraction using an alcohol/water mixture has been shown to not only extract soluble active ingredients but also increase the content of inert active ingredients in water, optimizing the extraction of fairly small amounts of active ingredients present in natural products, which can have a significant impact on biological activity [26]. An analysis of the main components of JGT in a previous report found that the main ingredients of JGT were oxypaeoniflorin, paeoniflorin, pentagalloylglucose, benzoyl paeoniflorin, liquiritin apioside, liquiritin, isoliquiritin, apioside, liquiritigenin, and glycyrrhizin. Among these, the relative amount of the compounds excluding albiflorin increased by 19–53% in JGTE compared to JGTW [7]. As such, it is likely that the content of various compounds contained in JGT increases in 30% EtOH extracts, leading to a combined effect on the anti-inflammatory activity in LPS-treated macrophages and MSU-induced gouty arthritis mouse model.

## 4. Materials and Methods

### 4.1. JGT Preparation

Paeoniae Radix and Glycyrrhizae Radix et Rhizoma were purchased from Omniherb (Daegu, Korea). According to the information contained in Donguibogam, Paeoniae Radix (100 g) and Glycyrrhizae Radix et Rhizoma (50 g), which are the chopped mixture materials of JGT, were prepared for extraction. Distilled water or 30% (*v*/*v*) ethanol was added to JGT, and the mixture was extracted for 1 h in a reflux extractor (MS-DM607; M-TOPS, Seoul, Korea). The extracts were then filtered, concentrated under reduced pressure in a rotary evaporator (N-1200A; Eyela, Tokyo, Japan) at 50 °C, and freeze-dried (FDU-2100; Eyela) at −80 °C for 72 h to obtain either a water extract (JGTW; yield 45.8%) or 30% (*v*/*v*) ethanol extract (JGTE; yield 39.6%).

### 4.2. 5-LOX and COX Assay

5-LOX activity was determined using a commercial 5-LOX inhibitor screening assay kit (Cayman Chem., Co., Ann Arbor, MI, USA, Cat #760700) according to the manufacturer’s protocol. COX inhibition was determined by an enzymatic colorimetric method using a commercial (ovine) COX inhibitor screening assay kit (Cayman Chem., Co., Cat #760111) according to the manufacturer’s protocol.

### 4.3. Cell Culture

The murine macrophage cell line RAW264.7 was obtained from the American Type Culture Collection (ATCC, Manassas, VA, USA). Cells were cultured in DMEM (Gibco Inc., Grand Island, NY, USA) supplemented with 5% heat-inactivated foetal bovine serum(Gibco Inc.), penicillin (100 U/mL), and streptomycin (100 μg/mL) and maintained in a 37 °C humidified incubator containing 5% CO_2_. After replacing the medium with serum-free DMEM, 0.5 μg/mL LPS (Sigma–Aldrich Chemical Co., St. Louis, MO, USA) was added with or without JGTW and JGTE (50, 100, and 200 µg/mL) for an additional 24 h to stimulate the cells.

### 4.4. Cytotoxicity Assay

Cell viability assays were performed to determine the cytotoxicity of JGT using 3-(4,5-dimethylthiazol-2-yl)-2,5-diphenyltetrazolium bromide (MTT) (Sigma–Aldrich Chemical Co.). Cells were plated in 96-well culture plates and incubated for 4 h with 100 μL of MTT dissolved in serum-free DMEM at a concentration of 0.5 mg/mL. The plates were removed from the incubator, and the formazan crystals were dissolved by the addition of 100 μL of dimethyl sulfoxide. The absorbance at 490 nm was measured using a microplate reader (Bio-Rad, Hercules, CA, USA) to measure cell viability. The absorbance was normalised to that of cells incubated in the control medium, which were considered 100% viable. The percentage of cell viability was calculated as follows: [Mean optical density (OD) in JGTW- and JGTE-treated cells/Mean OD in untreated cells × 100].

### 4.5. Real-Time PCR Analysis

Total RNA was isolated following the manufacturer’s protocol using a RNeasy Mini RNA Isolation Kit (QIAGEN, Hilden, Germany). cDNA was synthesised using 1 μg RNA and transcribed with a cDNA Synthesis Kit (BioRad, Hercules, CA, USA). Real-time PCR was performed with SYBR Green PCR Master Mix (Bio-Rad). The primer sequences are listed in Table 2. The mRNA levels of GAPDH were determined for the normalisation of the TNF-α, iNOS, COX-2, IL-6, and IL-1β mRNA expression values using CFX Manage^TM^ Software (Bio-Rad).

### 4.6. MSU Crystals-Induced Inflammation in Mice

Male C57BL6 mice (7 weeks) were purchased from Orient Bio (Seongnam, Korea). Mice were maintained in an air-conditioned room with a 12-h light/12-h dark cycle at a temperature of 22 ± 2 °C and humidity of 50 ± 10% with food and water available ad libitum. The experimental design was approved by the Committee on Animal Care of the KIOM, and all experiments were performed in accordance with committee guidelines (Approval No. 20-041, 26 March 2020). For gouty arthritis induction, MSU was synthesised as previously described [16]. After acclimatisation, C57BL6 male mice (8 weeks old, 20–22 g body weight) were divided into the following eight groups (*n* = 5/group): (1) control group; (2) MSU crystal-treated group; (3) 300 mg/kg MSU-JGTW group; (4) 200 mg/kg MSU-JGTW group; (5) 300 mg/kg MSU-JGTE group; (6) 200 mg/kg MSU-JGTE group; (7) 100 mg/kg MSU-JGTE group; (8) 1 mg/kg MSU-colchicine (Col) group. JGTW, JGTE, and Col were dispersed in 0.5% CMC and administered by oral gavage once daily for 4 days. One hour after drug treatment on day 1, gouty arthritis was induced by intradermal injection of 50 µL (4 mg) of MSU crystal suspension in PBS with 0.5% Tween 80 into the right paw; control (1) mice were administered PBS with 0.5% Tween 80.

### 4.7. Assessment of Inflammatory Paw Oedema and Pain

Inflammatory paw oedema was quantified at the end of the experimental period by measuring the thickness of the MSU-injected paw using a Vernier scale. The pain was assessed by examining changes in weight-bearing using a dynamic weight-bearing device (Bioseb, Boulogne, France), developed to measure the weight borne in each limb in freely moving animals [16,27]. The weight distribution ratio was calculated using the following equation: [weight on right hind limb/(weight on right hind limb + weight on left hind limb)] × 100.

### 4.8. Measurement of Inflammatory Mediators

NO production was analysed using the Griess Reagent System, according to the manufacturer’s instructions (Promega, Madison, WI, USA). The levels of IL-1β, IL-6, TNF-α, PGE2, and MPO were measured using ELISA kits from R&D Systems (Minneapolis, MN, USA) and MyBioSource (San Diego, CA, USA) according to the manufacturer’s protocol.

### 4.9. Statistical Analysis

Statistical analyses were conducted using Prism 7.0 (GraphPad Software, San Diego, CA, USA), and *p*-values < 0.05 were considered statistically significant. Values are expressed as mean ± standard error of the mean (SEM). The results were analysed using a one-way analysis of variance (ANOVA) followed by Dunnett’s tests for multiple comparisons or unpaired Student’s *t*-tests for two-group comparisons.

## 5. Conclusions

This study demonstrated that JGT exerted its anti-inflammatory activity via inhibition of inflammatory mediators (NO, iNOS, PGE2, and COX-2) and pro-inflammatory cytokines (IL-1β, TNF-α, and IL-6) in LPS-treated RAW264.7 cells. In addition, JGT efficiently downregulated MSU crystal-induced swelling and pain and contrasted inflammation by suppressing pro-inflammatory cytokines (IL-1β, TNF-α, and IL-6) and MPO activity in a gouty arthritis mouse model. These observations show that 30% ethanol (JGTE) is a good solvent for the extraction of JGT, which enhances anti-inflammatory efficacy even at reduced dosages compared with water extract (JGTW). These results may lead to further studies to determine whether JGTE has similar efficacy in clinical trials at dosages lower than JGTW.

## Figures and Tables

**Figure 1 ijms-21-09748-f001:**
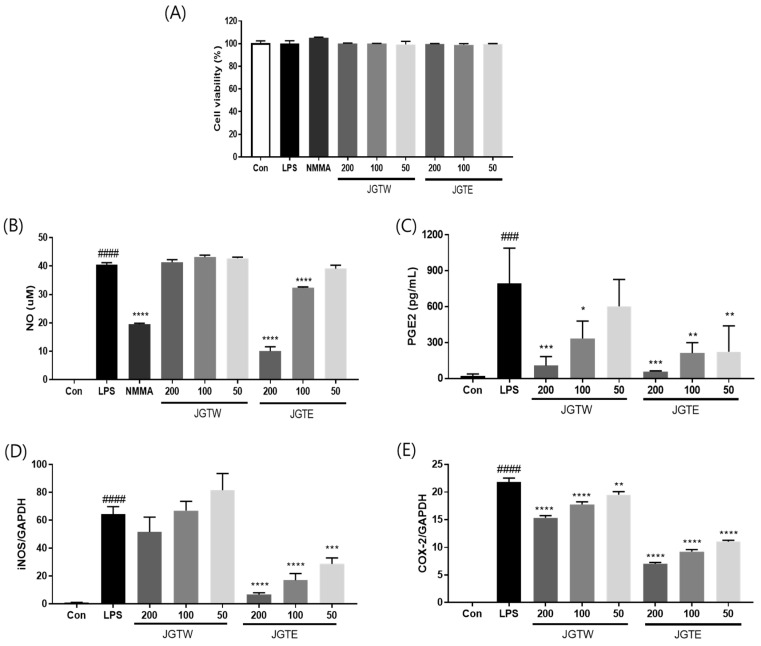
Effect of Jakyakgamcho-Tang (JGTW) and 30% EtOH Jakyakgamcho-Tang (JGTE) on viability, nitric oxide (NO), prostaglandin (PG) E2 production, and inducible nitric oxide/cyclooxygenase-2 (iNOS/COX-2) expression of lipopolysaccharide (LPS)-induced RAW264.7 macrophages. RAW264.7 cells were pre-treated with JGTW or JGTE (50, 100, and 200 µg/mL) for 2 h, and then stimulated with LPS (0.5 µg/mL) for 24 h. (**A**) Cell viability was measured using MTT (3-(4,5-dimethylthiazol-2-yl)-2,5-diphenyltetrazolium bromide) assay. (**B**) NO production was measured using a Griess reagent assay. L-NG-monomethyl arginine (L-NMMA) was used as a positive control. (**C**) PGE2 was measured using ELISA. The levels of (**D**) iNOS and (**E**) COX-2 were measured by real-time PCR. Values are expressed as means ± SD (*n* = 3). ### *p* < 0.001, #### *p* < 0.0001 vs. vehicle control cells. * *p* < 0.05, ** *p* < 0.01, *** *p* < 0.001, **** *p* < 0.0001 vs. LPS-induced cells.

**Figure 2 ijms-21-09748-f002:**
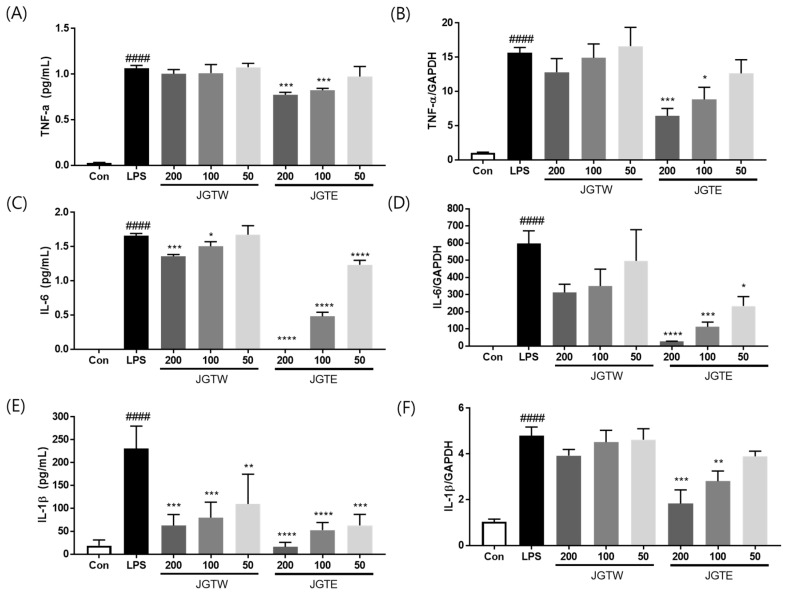
Effect of JGTW and JGTE on production of TNF-α, IL-6, and IL-1β in LPS-stimulated RAW264.7 macrophages by (**A**,**C**,**E**) ELISA and (**B**,**D**,**F**) real-time PCR. RAW264.7 cells were pre-treated with JGTW or JGTE (50, 100, and 200 µg/mL) for 2 h, and then stimulated with LPS (0.5 µg/mL) for 24 h. The values are expressed as the mean ± SD (*n* = 3). #### *p* < 0.0001 vs. vehicle control cells. * *p* < 0.05, ** *p* < 0.01, *** *p* < 0.001, **** *p* < 0.0001 vs. LPS-induced cells.

**Figure 3 ijms-21-09748-f003:**
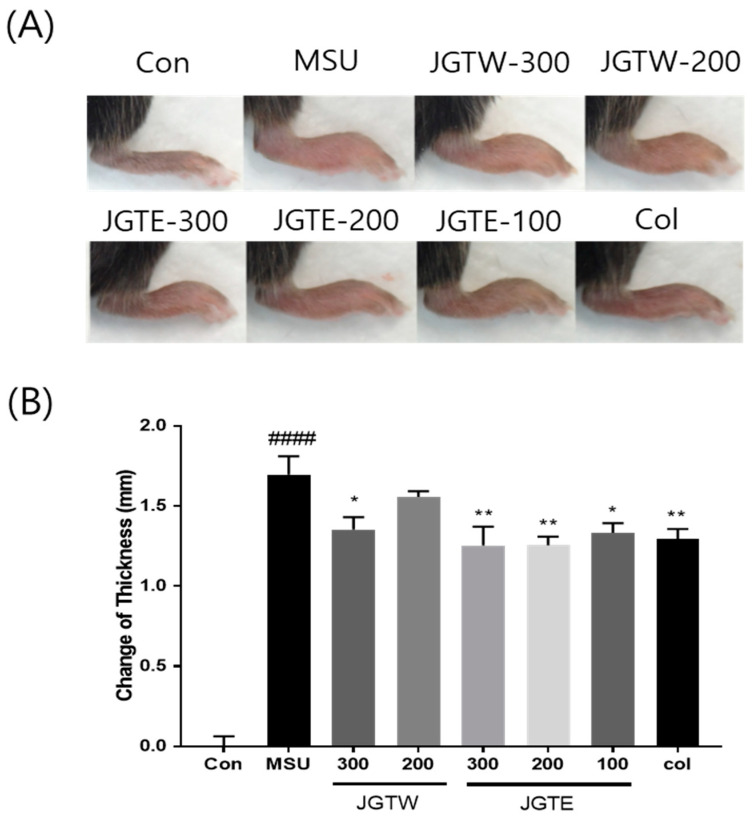
Effect of JGTW and JGTE on paw oedema in mice with monosodium urate (MSU) crystal-induced gouty arthritis. Con, control mice; MSU, MSU crystal-injected mice; JGTW, MSU-injected mice treated with JGTW; JGTE, MSU-injected mice treated with JGTE; Col, MSU-injected mice treated with 1 mg/kg of colchicine. (**A**) Representative images of the right leg from mice in each group are shown. (**B**) Measurement of the thickness of each mouse paw recorded at the end of the experimental period. Data are presented as the mean ± SEM (*n* = 5). #### *p* < 0.0001 vs. Control group and * *p* < 0.05, ** *p* < 0.01 vs. MSU group.

**Figure 4 ijms-21-09748-f004:**
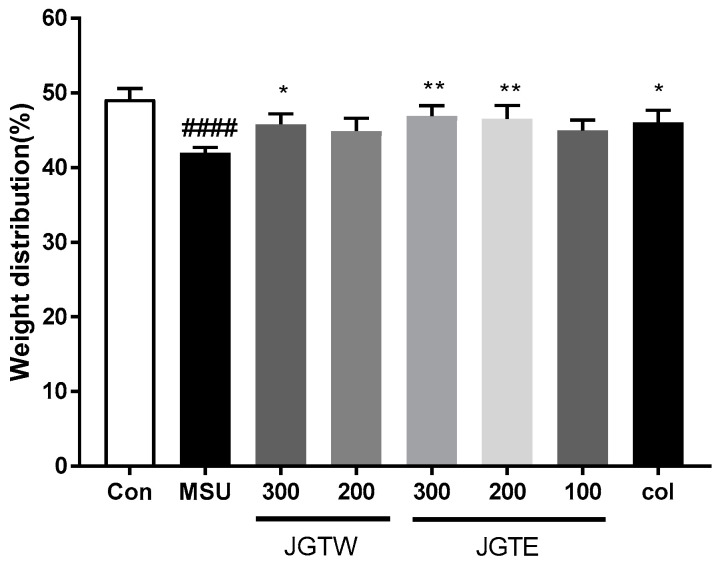
Effect of JGTW and JGTE on hind paw weight-bearing distribution in mice with MSU crystal-induced gouty arthritis. The weight-bearing distribution ratio was measured using a dynamic weight-bearing (DWB) device, compared to that of the MSU crystal-injected group. Con, control mice; MSU, MSU crystal-injected mice; JGTW, MSU-injected mice treated with JGTW; JGTE, MSU-injected mice treated with JGTE; Col, MSU-injected mice treated with 1 mg/kg of colchicine. Data are presented as the mean ± SEM (*n* = 5). #### *p* < 0.0001 vs. Control group and * *p* < 0.05, ** *p* < 0.01 vs. MSU group.

**Figure 5 ijms-21-09748-f005:**
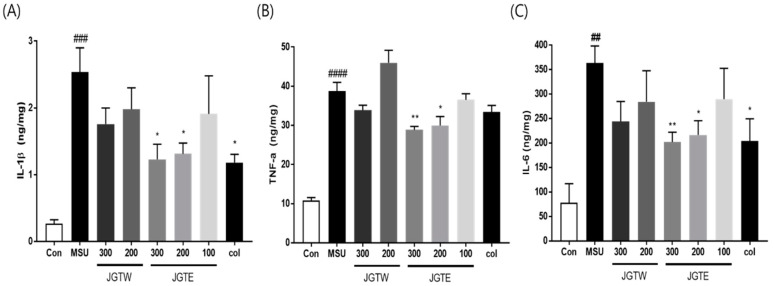
Effects of JGTW and JGTE on expression levels of proinflammatory cytokines in MSU crystal-injected paw tissue. Con, control mice; MSU, MSU crystal-injected mice; JGTW, MSU-injected mice treated with JGTW; JGTE, MSU-injected mice treated with JGTE; Col, MSU-injected mice treated with 1 mg/kg of colchicine. The levels of (**A**) interleukin (IL)-1β, (**B**) tumour necrosis factor (TNF)-α, and (**C**) IL-6 were measured by ELISA. Data are presented as the mean ± SEM (*n* = 5). ## *p* < 0.01, ### *p* < 0.001, #### *p* < 0.0001 vs. Control group and * *p* < 0.05, ** *p* < 0.01 vs. MSU group.

**Figure 6 ijms-21-09748-f006:**
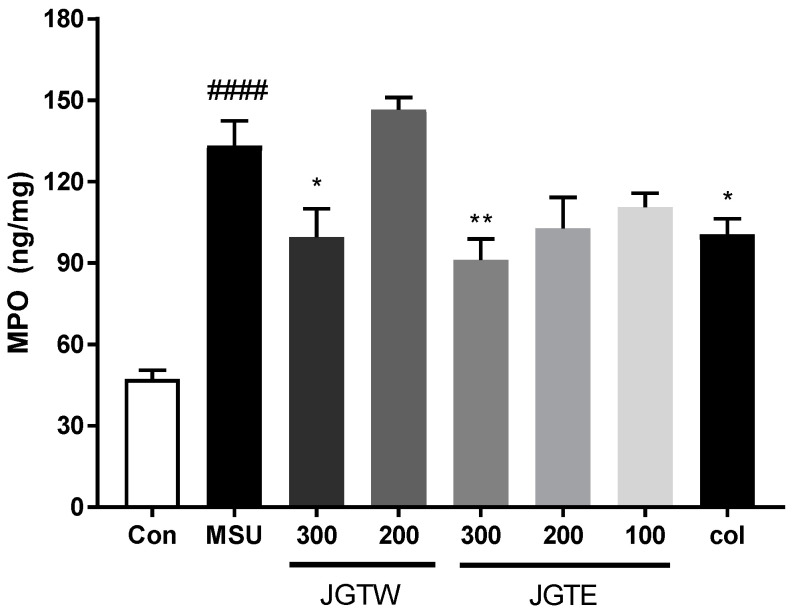
Effects of JGTW and JGTE on myeloperoxidase (MPO) activity in MSU crystal-injected paw tissue. Con, control mice; MSU, MSU crystal-injected mice; JGTW, MSU-injected mice treated with JGTW; JGTE, MSU-injected mice treated with JGTE; Col, MSU-injected mice treated with 1 mg/kg of colchicine. Data are presented as mean ± SEM (*n* = 5). #### *p* < 0.0001 vs. Control group and * *p* < 0.05, ** *p* < 0.01 vs. MSU group.

**Table 1 ijms-21-09748-t001:** The effect of water Jakyakgamcho-Tang (JGTW) and 30% EtOH Jakyakgamcho-Tang (JGTE) on 5-lipoxygenase (5-LOX), cyclooxygenase-1 (COX-1), and COX-2 enzymatic activity.

IC_50_	5-LOX	COX-1	COX-2
JGTW (μg/mL)	>500	219.47 ± 9.27	348.47 ± 0.83
JGTE (μg/mL)	74.63 ± 0.56	97.77 ± 13.86	114.16 ± 34.49

**Table 2 ijms-21-09748-t002:** Real-time PCR primer sequences.

Gene		Primer Sequence
IL-1β	Forward	5’- AGTGCAGCTGTCTAATGGGA-3’
	Reverse	5’- GCCCATCCTCTGTGACTCA-3’
IL-6	Forward	5’- TCAGAATTGCCATTGCACA-3’
	Reverse	5’- GTCGGAGGCTTAATTACACATG-3’
COX-2	Forward	5’- TGCATGTGGCTGTGGATGTCATCAA-3’
	Reverse	5’-CTCCTGCCCACTGAGTTCGTC-3’
TNF-α	Forward	5’- GCAGAGAGGTTGACTTTC-3’
	Reverse	5’- CTACTCCCAGGTTCTCTTCAA-3’
iNOS	Forward	5’-GTGTTCCACCAGGAGATGTTG-3’
	Reverse	5’-CTCCTGCCCACTGAGTTCGTC-3’
GAPDH	Forward	5’-CAAGAAGGTGGTGAAGCA-3’
	Reverse	5’-GGTGGAAGAGTGGGAGTT-3’

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
