# Peer review of "The Extraction Solvent Influences the Anti-Inflammatory Effects of Jakyakgamcho-Tang in Lipopolysaccharide-Stimulated Macrophages and Mice with Gouty Arthritis"

_ijms, 2020, doi:10.3390/ijms21249748_

Round 1

Reviewer 1 Report

The manuscript entitled “The extraction solvent influences the anti-inflammatory effects of Jakyakgamcho-tang in lipopolysaccharide-stimulated macrophages and mice with gouty arthritis” described the anti-inflammatory activity of Jakyakgamcho-tang (JGT) by evaluating the productions of inflammation cytokines in vitro and demonstrated the anti-gout effect in mice with monosodium urate induced gouty arthritis. Furthermore, authors prepared JGT by water extraction (JGTW) or by 30% ethanol extraction (JGTE). They compared anti-inflammatory effects of JGTW and JGTE, and concluded that JGTE was more effective than JGTW.  This is the first study that proved anti-gout effects of JGT in vivo. However, the anti-inflammatory activity of JGT is well demonstrated by using RAW 264.7 cells in many articles.

Here are some comments below:

  1. It is interesting to me that JGTE is more potent than JGTW. The traditional preparation of JGT is extracted by water. The chemical composition of JGTE is probably different from JGTW. Is there any differences of the chemical profiles of JGTW and JGTE? Besides, I am wondering the toxicity of JGTE. Are there any safety data of JGTE provided? Are there any examples supporting that extraction by 30% ethanol is more effective than extraction by water?
  2. There are some mistakes, such as the different size of letters in the line 212, should be corrected.

Author Response

  1. It is interesting to me that JGTE is more potent than JGTW. The traditional preparation of JGT is extracted by water. The chemical composition of JGTE is probably different from JGTW. Is there any differences of the chemical profiles of JGTW and JGTE? Besides, I am wondering the toxicity of JGTE. Are there any safety data of JGTE provided? Are there any examples supporting that extraction by 30% ethanol is more effective than extraction by water?

Answer: As mentioned in the 8th line of the introduction, we confirmed that the content of EtOH extract was increased by 19-53% compared to water extract through the HPLC analysis1). As mentioned in the 64th line of the introduction, JGT has traditionally been extracted with water, but recently, when developing pharmaceuticals and dietary supplements, a mixture of water and ethanol is frequently used, and the KFDA exempts or requires for use with a minimum toxicity test of less than 30% ETOH. After the treatment with JGTW and JGTE, no evidence of systemic adverse effects was observed in any study group. In normally, extraction using an alcohol/water mixture has been shown to not only extract soluble active ingredients, but also increase the content of inert active ingredients in water, optimizing the extraction of fairly small amounts of active ingredients present in natural products, which can have a significant impact on biological activity2). As such, it is likely that the content of various compounds contained in JGT increases in 30 % EtOH extracts, leading to a combined effect on the anti-inflammatory activity in LPS-treated macrophages and MSU-induced gouty arthritis mouse model.

  1. Comparison of Ingredient Quantities and Anti-Fatigue Effects of Jakyakgamcho-         Tang according to Extraction Solvent. The Korea Journal of Herbology 2020, 35, (2), 31-38).  
  2. 2) Phytochemicals: Extraction, Isolation, and Identification of Bioactive Compounds from Plant Extracts. Plants (Basel) 2017, 6, (4).
  3. There are some mistakes, such as the different size of letters in the line 212, should be corrected.

Answer: I changed font size.

Reviewer 2 Report

This article discusses the anti-inflammatory activity of JGT via inhibition of inflammatory mediators (NO, iNOS, PGE2, and COX-2) and pro-inflammatory cytokines (IL-1β, TNF-α, and IL-6) in LPS-treated RAW264.7 cells. In addition, JGT efficiently downregulated MSU 314 crystal-induced swelling and pain, and contrasted inflammation by suppressing proinflammatory cytokines and MPO activity in a gouty arthritis mouse model. Finally, the authors' data show that JGTE is a good solvent for the extraction of JGT, which enhances anti-inflammatory efficacy even at reduced dosages compared with JGTW. This is an interesting study, but the overall impact of the study is limited for the overall descriptive character of the work and for to the lack of specificity of some methods used.

I would recommend to consider this for publishing after major revisions and clarifications.

  • Inflammation is a feature present in gouty arthritis, and in vitro cell treatment with LPS certainly represents an excellent model for analyzing the expression and levels of specific inflammatory markers. I don't understand why they didn't use the RAW264.7 macrophage cell line treated with MSU crystals as a gouty model in vitro. I believe that the authors should include such experiments, considering that several experimental evidences have reported that the treatment of macrophages with MSU induces inflammation and increase of the same cytokines analyzed by the authors.
  • Furthermore, both COX and iNOS are regulated by the transcription factor NFkB. Expression analysis by Real-Time PCR shows that, in an insignificant way, both JGTW and JGTE reduce the expression of COX in a dose dependent manner, and only JCTE of iNOS. The authors would also have analyzed the NfkB transcription factor. But, a part from that, could the authors better explain whether JGTW and JGTE act primarily as inhibitors of COX-2 (IC5o) enzyme activity or whether they primarily inhibit the expression of this inducible protein?
  • In the figures it is not understood that the LPS + JGT co-treatment was carried out. Authors are requested to improve this graphic aspect.

Author Response

  1. Inflammation is a feature present in gouty arthritis, and in vitro cell treatment with LPS certainly represents an excellent model for analyzing the expression and levels of specific inflammatory markers. I don't understand why they didn't use the RAW264.7 macrophage cell line treated with MSU crystals as a gouty model in vitro. I believe that the authors should include such experiments, considering that several experimental evidences have reported that the treatment of macrophages with MSU induces inflammation and increase of the same cytokines analyzed by the authors.

Answer: Activation of macrophages by LPS causes production of inflammatory mediators (NO, iNOS, PGE2, and COX-2) and pro-inflammatory cytokines (TNF-α, IL-1β, and IL-6), which contribute to the progression of several inflammatory diseases. To identify herbal plants with anti-inflammatory effects, we screened herbal medicinal plant formula extracts and found JGT. So, we applied to an animal model of gouty arthritis characterized by an intense inflammatory process and pain. In vitro using the RAW264.7 macrophage cell line treated with MSU crystals as a gout model, it was difficult to confirm the change of production of inflammatory mediators (NO, iNOS, PGE2, and COX-2) and pro-inflammatory cytokines (IL-1β, and IL-6) by ELISA. Because there was no significant difference between the MSU-treated group and the Con group, it was difficult to confirm the efficacy of the JGTW and JGTE.

  1. Furthermore, both COX and iNOS are regulated by the transcription factor NFkB. Expression analysis by Real-Time PCR shows that, in an insignificant way, both JGTW and JGTE reduce the expression of COX in a dose dependent manner, and only JCTE of iNOS. The authors would also have analyzed the NF-kB transcription factor. But, a part from that, could the authors better explain whether JGTW and JGTE act primarily as inhibitors of COX-2 (IC5o) enzyme activity or whether they primarily inhibit the expression of this inducible protein?

Answer: In line 196, the IC50 of JGTE for COX-1 and COX-2 (97.77 μg/mL and 114.16 μg/mL) was 2.24-fold and 3.05-fold lower than that of JGTW (219.47 μg/mL and 348.47 μg/mL), respectively. Therefore, JGTE acts as a more efficient inhibitor of cox-1 and cox-2 enzyme activity than JGTW. And line 185-192, both JGTW and JGTE efficiently reduced the production of PGE2 via suppression of COX-2 expression. JGTE reduced NO production in a concentration-dependent manner, which correlated well with the dose-dependent decrease in iNOS mRNA levels. However, JGTW slightly decreased, although not significantly, and no change of NO inhibition.

  1. In the figures it is not understood that the LPS + JGT co-treatment was carried out. Authors are requested to improve this graphic aspect.

Answer: I added this sentence as follows in figure legend 1 and 2.

"RAW264.7 cells were pre-treated with JGTW or JGTE (50, 100, and 200 µg/mL) for 2h, and then stimulated with LPS (0.5 µg/ mL) for 24h."

Round 2

Reviewer 2 Report

Dear authors, the work is now clearly displayed, however please review the graphics in table 1.